# Synergistic Effect of Squalene and Hydroxytyrosol on Highly Invasive MDA-MB-231 Breast Cancer Cells

**DOI:** 10.3390/nu14020255

**Published:** 2022-01-07

**Authors:** Cristina Sánchez-Quesada, Francisco Gutiérrez-Santiago, Carmen Rodríguez-García, José J. Gaforio

**Affiliations:** 1Department of Health Sciences, Faculty of Experimental Sciences, University of Jaen, 23071 Jaen, Spain; csquesad@ujaen.es (C.S.-Q.); fgutierr@ujaen.es (F.G.-S.); crgarcia@ujaen.es (C.R.-G.); 2University Institute of Research in Olive Groves and Olive Oils, University of Jaen, Campus las Lagunillas s/n, 23071 Jaen, Spain; 3Agri-Food Campus of International Excellence (ceiA3), 14071 Cordoba, Spain; 4CIBER Epidemiología y Salud Pública (CIBER-ESP), Instituto de Salud Carlos III, 28029 Madrid, Spain

**Keywords:** virgin olive oils, breast cancer, antitumor, proliferation, apoptosis, DNA damage, comet assay

## Abstract

Several studies relate Mediterranean diet and virgin olive oil (VOO) intake with lower risk of several chronic diseases, including breast cancer. Many of them described antitumor properties of isolated minor compounds present in VOO, but beneficial properties of VOO arise from the effects of all its compounds acting together. The aim of the present study was to test the antitumor effects of two minor compounds from VOO (hydroxytyrosol (HT) and squalene (SQ)) on highly metastatic human breast tumor cells (MDA-MB-231) when acting in combination. Both isolated compounds were previously analyzed without showing any antitumoral effect on highly invasive MDA-MB-231 breast cancer cells, but the present results show that HT at 100 µM, combined with different concentrations of SQ, could exert antitumor effects. When they are combined, HT and SQ are able to inhibit cell proliferation, promoting apoptosis and DNA damage in metastatic breast cancer cells. Therefore, our results suggest that the health-promoting properties of VOO may be due, at least in part, to the combined action of these two minor compounds.

## 1. Introduction

Among all types of cancer, breast cancer is one of the most common causes of mortality among women in developed countries [1]. Many environmental factors, such as diet, can affect breast cancer development [2]. In this context, the Mediterranean diet pattern is associated with low incidence of breast cancer [3]. Several authors associate these health effects with virgin olive oil (VOO), the main source of fat in the Mediterranean diet [4,5]. Growing scientific evidence suggests that healthy properties of VOO reside in their minority compounds such as polyphenols, triterpenes, tocopherols and lignans [6,7]. Among them, polyphenols were found to be related to the prevention of diabetes, neurological and cardiovascular diseases, and cancer [8].

Hydroxytyrosol (HT) is one of the main phenolic compounds present in VOO, although it appears in a smaller amount than tyrosol [9]. Several studies have demonstrated biological activities of HT, both in vitro and in vivo. HT is a strong antioxidant that exerts a scavenger function of oxygen and nitrogen free radicals [10]. Furthermore, HT protects DNA against oxidative damage in neuronal hybridoma cells [11]. It also has anti-inflammatory and analgesic properties [12], and exerts antitumor activities in colon and breast cancer cells [13,14], with pro-apoptotic effects through the modulation of gene expression [15]. The mechanism by which HT exerts its effects on cancer cells is not clear, but it could produce a reduction in Pin1 levels that causes the translocation of cyclin D1 to the cytoplasm, where it is degraded. Cyclin D1 is a necessary protein to the G1/S cell cycle transition and, consequently, for tumor cell growth [16]. Furthermore, in a previous study conducted by our group, HT was able to protect DNA in breast normal cells in vitro [17], pointing it out as a protector against the development of breast cancer.

Squalene (SQ) is the main triterpene hydrocarbon present in VOO. VOO is the highest source of SQ compared to other vegetable fats [18]. SQ has a lot of interesting activities such as antioxidant and antitumor effects [19]. Among other bioactivities, SQ exerts a potent inhibition of aberrant hyperproliferation in mammary epithelial cells [20].

SQ is a metabolite involved in the biosynthesis of cholesterol and it is oxidized by monooxygenase in the early stages of metabolism. However, its monooxygenase activity is highly suppressed by the accumulation of cholesterol [21]. Therefore, when squalene is ingested by diet, squalene-derived sterols accumulate in the cells and β-hydroxy-β-methylglutaryl-CoA reductase (HMG-CoA reductase) is inhibited [21]. Furthermore, recent evidence has shown that squalene excess leads to an accumulation of farnesyl pyrophosphate (FFP), which could be related to the suppression of carcinogenesis [22,23]. In fact, dietary ingestion of SQ can inhibit colon cancer development in vivo [24]. On the other hand, it was suggested that SQ enhances the action of the immune system against tumors [25] and protects breast cells against the accumulation of mutagenic lesions in their DNA, according to a previous work published by our group [26].

The bioprotective and antitumoral activities of VOO have been linked to its contents of minor compounds. Several studies remarked on different bioactivity properties of isolated VOO compounds [6,9,10,11,12,13,14,15,16,17,18,19,20,21,22,23,24,25,26], but VOO’s health-promoting properties could be attributed to synergistic effects of the nutrients present in it, such as minor compounds and fatty acids. HT and SQ are two of the minor compounds of VOO and both isolated compounds have been previously described to possess a preventive role in breast epithelial cells but without significant antitumoral activity in highly invasive breast cancer cells [17,26]. We hypothesized that the well-known chemopreventive effects of VOO against breast cancer may be due not to an isolated compound, but the synergistic activity of several minor compounds acting together. The present study attempts to demonstrate the possible synergistic antitumoral effect of these two minority compounds in breast cancer cells.

## 2. Materials and Methods

### 2.1. Materials

TrypLE Express and Minimum Essential Medium with Eagle’s salts (MEM) were obtained from Gibco^®^ Life Technologies Ltd. (Paisley, UK). Fetal bovine serum (FBS) was purchased from de PAA Laboratories GmbH (Pasching, Austria). The following products were obtained from Sigma-Aldrich Co. (St Louis, MO, USA): 2,2′-Azino-bis(3-ethylbenzthiazoline-6-sulphonic acid) diammonium salt tablets (CAS 30931-67-0 (ABTS); (S)-(+)-camptothecin (CAS 7689-03-4 (CPT)) purity ≥ 90%; 2,2-Diphenyl-1-picrylhydrazyl (DPPH) purity ~90%; 6-hydroxy-2,5,7,8-tetramethylchroman-2-carboxylic acid (Trolox™ CAS 53188-07-1 (TR)) purity ≥ 97%; Phosphate Buffer Saline (PBS); DL-all-rac-α-tocopherol (Vitamin E-CAS 10191-41-0 (TOC)) purity ≥ 96%; Dimethyl sulfoxide (DMSO); 2′,7′-dichlorofluorescein diacetate (DCFH-DA); Non-Essential Amino Acids mixture 100× (NEAA); Sodium Pyruvate; Hepes Buffer; 2,6,10,15,19,23-hexamethyl-2,6,10,14,18,20-tetracosahexane (Squalene CAS 111-02-4). 2-(3,4-dihydroxyphenyl)-ethanol (Hydroxytyrosol CAS 10597-60-1) was acquired from Cayman Chemical (Ann Arbor, MI, USA). Binding Buffer and FITC-conjugated Annexin V were purchased from Miltenyi Biotec (Cologne, Germany). The PI/RNase Staining Buffer kit was purchased from BD Biosciences Pharmigen (San Diego, CA, USA). Phosphate Buffer Saline (PBS 1x) (Ca^2+^/Mg^2+^ free) was obtained from (Gatersleben, Germany). CellTiter-Blue^®^ Cell Viability Assay was obtained from Promega Corporation (Madison, WI, USA).

### 2.2. Scavenging Radical Activity Estimation by DPPH Assay

Antioxidant activity of the HT–SQ combination (HT–SQ) against DPPH radical was performed as previously described [27], incorporating certain changes. An ethanolic solution 100 μM of DPPH was mixed in 96-well plates with various solutions of antioxidant standard (α-tocopherol) or HT–SQ in ratios of 0.03, 0.13, 0.5 and 2 moles of antioxidant/moles of DPPH. Absorbance at 520 nm was read every 5 min during 2 h using a microplate reader (TECAN GENios Plus, Tecan Group Ltd., Männedorf, Switzerland). Triplicate measurements were conducted in 3 independent experiments. Furthermore, DPPH samples were measured as blank controls. The radical scavenging activity percentage (% RSA) was estimated using the formula below:% Radical Scavenging Activity = 100 (A_C(0)_ − A_A(t)_)/A_C(0)_
A_C(0)_ = Control absorbance at t_0_
A_A(t)_ = HT–SQ/standard absorbance at t_60_

### 2.3. ABTS Radical Scavenging Assay

ABTS cation radical scavenging activity was analyzed according to a previously described protocol [28], adding certain modifications. ABTS radical cations (ABTS^●+^) were generated through the reaction of 2.45 mM K_2_S_2_O_8_ with 7 mM ABTS over 16 h at room temperature (RT) in darkness. ABTS^●+^ obtained was diluted in ultrapure water (UPW) until absorbance ranged between 0.7 and 1 at 734 nm. Antioxidant standard (Ethanol (EtOH) solutions of Trolox^TM^) were diluted in UPW to achieve concentrations of 50 to 800 μM. HT–SQ were diluted in UPW to reach concentrations between 0.01 and 100 μM. Trolox^TM^, UPW (blank) or EtOH control (8%) were added in a flat-bottom 96-well plate. Different HT–SQ concentrations were added in the 96-well plate. Reactions were initiated by adding ABTS^●+^. Immediately, absorbance was measured at 734 nm every 5 min for 2 h at 30 °C with a microplate reader. Triplicate measurements were conducted in 3 independent experiments. % RSA was calculated as previously described (at t_60_).

### 2.4. Cell Culture

MDA-MB-231 cells (highly invasive and triple-negative human breast cancer cells) were purchased from the American Type Culture Collection (ATCC, Rockville, MD, USA). Cells were cultured in monolayer cultures in MEM supplemented with 1% NEAA, 1% Sodium Pyruvate, 1% HEPES Buffer and 10% FBS. Cells were subcultured using TrypLE Express solution and were maintained at 37 °C in a humidified atmosphere with 5% CO_2_. For all experiments, cells in the exponential growth phase were used. Different combinations of HT–SQ were tested for DPPH, ABTS, cytotoxicity and proliferation assays: HT ranged from 0.01 to 100 μM combined with SQ from 0.01 to 100 μM.

### 2.5. Cytotoxicity Assay

Cell viability was tested after treatment of MDA-MB-231 with different concentrations of HT–SQ using the CellTiter-Blue^®^ Cell Viability Assay based on the manufacturer’s specifications with certain amendments. Briefly, a total of 5 × 10^3^/well cells in 100 μL were cultured in 96-well plates. After overnight (O/N) incubation to ensure cell adhesion, cells were treated with increasing HT–SQ combinations (0.01 to 100 μM) for a further 24 h. Thereafter, cells were incubated for 3 h in darkness at 37 °C in 5% CO_2_ with CellTiter-Blue Cell Viability. Cells were treated with EtOH and DMSO alone and a combination of both as vehicle control. Relative fluorescence was measured in a plate reader (Ex. Λ_485_/Em. Λ_595_, Gain 60). All treatment combinations were conducted in triplicate in 3 independent experiments. Cell viability was estimated using the formula:Cell viability percentage = [A (treated cells)/A(control)] × 100
A = relative fluorescence units of each sample

### 2.6. Cell Proliferation Assay

Cell proliferation was conducted using a CellTiter-Blue Cell Viability Assay based on the manufacturer’s specifications, with some adaptations. It was measured as cell growth of untreated controls versus treated cells. Briefly, a total of 1 × 10^3^/well cells in 100 μL were cultured into 96-well plates. After O/N incubation to ensure cell fixation, cells were treated with increasing combinations of HT–SQ, from 0.01 to 100 μM. After 24 h of treatment, fresh medium was added and cells were incubated for another 24 h. Then, cells were incubated for 3 h in darkness at 37 °C in 5% CO_2_ with the CellTiter-Blue Cell Viability Assay kit and relative fluorescence units were measured in a plate reader (Ex. λ_485_/Em. λ_595_, Gain 60). Measurements were repeated after 72 and 96 h of treatments. As vehicle control, cells were treated with EtOH and DMSO alone and a combination of both. All treatment combinations were conducted in triplicate in 3 independent experiments. Cell viability was estimated with the following equation:Cell viability percentage = [A (treated cells)/A(control)] × 100
A = relative fluorescence units of each sample

### 2.7. Cell Cycle Assay

A total of 5 × 10^4^/well cells in 1 mL were cultured into 12-well plates and incubated O/N to ensure cell attachment. Afterward, cells were treated with 100 μM of HT combined with a range of SQ, from 0.01 to 100 μM, for 48 h. Thereafter, cells were harvested with TrypLE Express and washed with cold 1× PBS (300× *g*, 10 min at 4 °C). Eventually, cells were fixed in cold 70% EtOH and kept at −20 °C for at least 24 h. After propidium iodide labeling (PI/RNase Staining Buffer), cells were analyzed by flow cytometry (MACSQuant^®^ Analyzer 10, Miltenyi Biotec, Bergisch Gladbach, Germany). The MACSQuantify Software program was used for calculating the % of cells in different phases (G0/G1, S and G2/M). Each experiment was run independently 3 times.

### 2.8. Analysis of Apoptosis

Apoptotic cell percentage was assessed by a double staining assay with PI and FITC-conjugated Annexin V. A total of 5 × 10^4^/well cells in 1 mL were cultured in 12-well plates and incubated O/N to ensure attachment of cells. Afterward, cells were treated with 100 μM of HT combined with a range of SQ, from 0.01 to 100 μM, for 48 h. Thereafter, cells were harvested with TrypLE Express, centrifuged (300× *g*; 10 min at 4 °C) and resuspended in 100 μL of Annexin Binding Buffer 1x. Cells were stained with PI and Annexin V-FITC solution; then, they were gently vortexed and incubated for 15 min at RT in darkness prior to flow cytometry analysis. Cells were treated with camptothecin (CPT) as positive control. Every experiment was repeated independently 3 times.

### 2.9. Detection of Intracellular Reactive Oxygen Species

Intracellular reactive oxygen species (ROS) levels were evaluated using a cell-permeable fluorescent probe, 2′,7′-dichlorofluorescein diacetate (DCFH-DA), as previously described by Wang and Joseph [29], adding some modifications. Briefly, 3.5 × 10^3^/well cells in 100 μL were cultured in 96-well plates. After O/N incubation to allow cell attachment, cells were treated with 100 μM of HT combined with a range of SQ, from 0.01 to 100 μM, for 48 h. Thereafter, DCFH-DA (100 μM) was added during 30 min at 37 °C with 5% CO_2_. Fluorescence was measured in a plate reader during 30 min (Ex. λ485/Em. λ535, Gain 60). Intracellular ROS levels were calculated using the following formula:F = [(F(t = 30 min) − F(t = 0 min))/F(t = 0 min) × 100]

In culture cells, the addition of H_2_O_2_ was shown to increase oxidative stress and to damage DNA directly [30]. Therefore, 400 μM H_2_O_2_ was added 30 min prior to fluorescence quantification to assess the protective capacity of HT–SQ treatments against oxidative stress induction.

All tests were performed six-fold for each experimental situation, and each experiment was repeated 3 times. Experiments were carried out using iron-free media (MEM).

### 2.10. Alkaline Single-Cell Gel Electrophoresis (Comet Assay)

Cells were cultured into a 12-well plate (1 × 10^5^ cells/well) and incubated O/N to reach cell attachment. Cells were treated with different HT–SQ combinations. Then, cells were scraped and washed twice with cold 1X PBS (300× *g* 10 min, 4 °C). They were resuspended in 1 mL of cold 1x PBS. Cells were treated with 50 µM H_2_O_2_ for 10 min at 4 °C to assess the ability of HT (100 μM)-SQ (100, 10 and 1 μM) in protecting against oxidative DNA damage. Then, the comet assay was conducted as described in Warleta et al. [17]. Each experiment was repeated independently at least 3 times.

DNA strand breaks were screened using the Komet 5.5 software package (Kinetic Imaging Ltd., Liverpool, UK) in a fluorescence microscope (Zeiss Axiovert 200) equipped with a Luca EMCCD camera (Andor Technology, Belfast, UK) (Ex. 494 nm/Em; 521 nm wavelength). We randomly characterized 25 cell images per sample at a magnification of 20×. Relative fluorescence between head and tail, determined through the olive tail moment (Olive_TM), was used to determine DNA damage. Olive_TM is defined as the product of the Tail Moment Length and the fraction of DNA in the tail.
Olive_TM = [(tail (mean) − head (mean)) × tail (% DNA)]/100

### 2.11. Statistical Analysis

The results are shown as the mean of 3 independent experiments (±SEM), and are expressed as a relative percentage of untreated control (set as 100%). Statistical analysis was conducted using a one-way analysis of variance (ANOVA) followed by Fisher’s LSD test with the STATGRAPHICS Centurion XVI software (version 16.0.10) (Statpoint Technologies, Inc., Warrenton, VA, USA). *p* values < 0.05 were considered statistically significant.

## 3. Results

### 3.1. Estimation of Radical Scavenging Activity by the DPPH Test

The antiradical activity of the HT and SQ combinations, measured by DPPH assay, showed that HT at 2 mol ratio combined with all ratios of SQ (2, 0.5, 0.13 and 0.03 mol ratio) exhibited antioxidant activity (RSA > 40% in the four combinations) (Table 1). This antioxidant capacity was not observed when HT was below 2 mol ratio (data not shown). α-tocopherol was used as an antioxidant standard control.

### 3.2. Radical Scavenging Activity by the ABTS Assay

The ABTS antiradical assay showed that HT at 100 μM, combined with all concentrations of SQ (from 0.01 to 100 μM), exhibited a high scavenging activity (RSA > 60% at five combinations) (Table 2). HT at 10 μM combined with all the concentrations of SQ showed weak antiradical activity. HT below 10 μM combined with SQ did not present scavenging activity (data not shown).

### 3.3. Cytotoxicity Assay

To analyze the potential cytotoxic effects of HT–SQ, cells were treated with combinations of both HT and SQ, from 0.01 μM to 100 μM, for 24 h. The results are expressed as % of cell survival with respect to the untreated control, which was set as 100%. None of the tested combinations promoted cell death with statistical differences compared to the untreated control (Figure 1).

### 3.4. Cell Proliferation Assay

The proliferation of MDA-MB-231 cells was determined after treatment of HT–SQ from 0.01 to 100 μM for 24 h, and was measured again after another 24 h with fresh medium. After each 24 h, proliferation was measured in a microplate reader. The results are expressed as % of cell survival concerning the untreated control that was set as 100% (Figure 2). After 48 h, inhibition of the cell survival was observed with HT at 100 μM and all SQ concentrations assayed (Figure 2a). The same antiproliferative effect was maintained by these combinations at 72 h (Figure 2b) and after 96 h (Figure 2c) with statistical differences.

### 3.5. Cell Cycle Assay

Due to the results observed during the proliferation assay, MDA-MB-231 cells were treated with HT at 100 μM combined with SQ (from 0.01 to 100 μM) for 24 h and fresh medium was added for another 24 h. The results are expressed as cell percentages in the different phases of the cell cycle. None of the selected combinations produced cell cycle alterations in a statistically significant way (Figure 3). However, all the tested combinations produced an increment of the percentage of cells in the S phase. They also diminished (without statistical significance) the percentage of cells in the G0/G1 phase.

### 3.6. Analysis of Apoptosis

The percentages of living, apoptotic, and necrotic cells are represented with respect to the total, which was set as 100%. HT at 100 μM combined with SQ (from 0.01 to 100 μM) for 24 h plus 24 h of fresh medium were analyzed by flow cytometry. HT at 100 μM combined with all concentrations of SQ produced apoptosis in MDA-MB-231 cells (Table 3). HT at 100 μM combined with SQ at 10 and 100 μM produced apoptotic cells with statistical significance.

### 3.7. Detection of Intracellular Reactive Oxygen Species

Cells treated for 24 h with HT combined with SQ (from 0.01 to 100 μM), followed by another 24 h with fresh medium, were analyzed to determine ROS levels. None of the selected combinations decreased the ROS levels in MDA-MB-231 cells (Figure 4a).

H_2_O_2_ was added prior to fluorescence measurement to induce intracellular oxidative stress. To investigate the in vitro preventive effect of HT–SQ against H_2_O_2_ oxidative injury, intracellular levels of ROS were measured in cells that were previously treated with HT at 100 μM combined with all concentrations of SQ (from 0.01 to 100 μM) for 24 h followed by another 24 h with fresh medium. The results showed that at the lowest SQ concentration (0.01 μM), ROS levels were decreased (~50%). At higher SQ concentrations, levels of ROS increased in a dose-dependent manner (Figure 4b).

### 3.8. Analysis of DNA Damage

DNA damage in MDA-MB-231 was measured by unicellular alkaline electrophoresis after HT (100 μM)-SQ (100, 10 and 1 μM) treatments for 24 h followed by another 24 h in fresh medium. Due to apoptosis results, only three different concentrations of SQ were used to analyze the formation of possible DNA breaks. As Table 4 shows, HT–SQ treatment did not damage DNA, except for HT (100 μM)-SQ (1 μM), which promoted about 20% more damage to DNA than observed in control. After H_2_O_2_ burst, all assayed treatments promoted the prevention of DNA damage (Table 4).

## 4. Discussion

There is now scientific evidence linking the Mediterranean dietary pattern with the low incidence and prevalence of some chronic diseases such as certain types of cancer. VOO is the main source of fat consumed in this dietary pattern, and its regular intake reduces the incidence of some illnesses, such as cardiovascular diseases and a wide variety of cancers [5]. The minor compounds present in VOO could have a key role in its healthy properties [4,5,6,7,8,9,31,32,33,34]. However, most of these studies are focused mainly on the health benefits of isolated compounds from VOO. In the present article, we found that HT and SQ, both minor compounds present in VOO, have antitumor effects on highly invasive breast cancer cells only when they act in combination, but not when they act independently. HT is one of the main phenols present in VOO with interesting health benefits [11,12,13,14]. On the other hand, SQ, another important compound present in VOOs and olives, also has antitumor properties against several types of cancer, such as skin, breast and colon cancer [18,19,20,21,22,23,24]. Interestingly, both isolated compounds did not promote the inhibition of highly invasive breast cells in previous published works [17,26], but together at certain concentrations, appeared to promote the inhibition of breast cancer cells’ growth and apoptosis, as well as increasing DNA damage.

The combination of HT and SQ showed high radical scavenging activity in the DPPH and ABTS assays. Warleta et al. [26] showed that SQ does not present any radical scavenging activity in the DPPH and ABTS tests. On the contrary, HT has been reported to be a potent scavenger of free radicals [10,16]; for this reason, the antioxidant activity observed in this study could be due to the antioxidant activity per se of HT (Table 1 and Table 2). Further, the higher free radical scavenger activity occurred when HT was at the highest concentration used in both assays (DPPH and ABTS) regardless of the concentration of SQ, which reinforces the idea that HT is the compound responsible for this antioxidant effect.

This antioxidant effect of both compounds in combination could be related to oxidative radical accumulation that collaborates with breast neoplastic transformation. In fact, reactive oxygen species (ROS) play a crucial role in tumor progression [35]. In this context, SQ is a potent intracellular antioxidant [36]. An experimental study showed that SQ is able to reduce basal and H_2_O_2_-induced ROS levels in human breast MCF10A cells, but not in human breast cancer MCF-7 cells and MDA-MB-231 cells [26]. Unless SQ could have a protective activity against tumor development, this “selective antioxidant sensitivity” of SQ remains unknown. Moreover, HT is a well-known antioxidant phenol present in VOO but it is not able to reduce basal ROS levels in MDA-MB-231 breast cancer cells. Indeed, our group previously observed that in conditions of H_2_O_2_-induced ROS levels, it reduced ROS levels in a dose-dependent manner in these cancer cells [17]. According to that, our results showed that the combination of both HT and SQ did not reduce basal ROS levels in MDA-MB-231 cells; furthermore, when H_2_O_2_ was added, HT at 100 μM combined with SQ (from 0.01 to 100 μM) reduced ROS levels.

Due to the above, once a high-stress microenvironment is present in breast cancer, a combination of both compounds is not able to induce higher ROS levels, which could promote the inhibition of breast cancer growth. It is well known that the formation of neoplasms promotes a higher stress microenvironment than that observed in normal states, and cancer cells are able to adapt their metabolism to the new stress conditions [35]. However, once they are adapted, a change in ROS levels (a small enhancement or decrease) could be derived from the inhibition of cancer growth, due to the lack of oxidative stress enzymes [35]. Interestingly, combination of HT–SQ is able to increase DNA damage in normal conditions (Table 4), which suggests that the combination of these two compounds could inhibit breast cancer growth, but with another pathway that is different from ROS management (Figure 4a). However, once a stress condition is induced in these cells (Table 4), the HT–SQ combination is able to reduce DNA damage, causing them to lose their antitumoral effect.

Indeed, an inhibition of proliferation was observed with HT–SQ at certain concentrations (Figure 2) as well as an apoptosis activity (Table 3), which reinforces the idea that the combination of both HT–SQ could be a useful tool to treat or even prevent some breast cancers. These HT–SQ effects were not observed in our previous works [17,26], where the research was focused on their antitumoral effects as isolated compounds.

The antiproliferative activity of HT was previously described in human colon adenocarcinoma cells by Corona et al. [13]. HT exerts its antiproliferative effects through the inhibition of the 1/2 phosphorylation of extracellular signal-regulated kinase (ERK) and a decrease in cyclin D1 expression [13]. This interesting activity of HT also was observed in human breast cancer cells. HT was capable of inhibiting cell proliferation in human breast cancer MCF-7 cells [16,37,38]. Extracts from dry olive mill residue with increasing contents of HT were a strong cell growth inhibitor [39]. We evaluated the antiproliferative effect of HT in MDA-MB-231 cells, but HT was found to only have a slight antiproliferative activity at 200 μM [17]. In contrast, SQ does not have any antiproliferative effect in MDA-MB-231 cells [26]. In the present work, we demonstrated that if HT was combined with SQ, the antiproliferative activity of HT appears at a lower concentration (100 μM) in these highly invasive breast cancer cells, but in combination with SQ.

The pro-apoptotic activity of HT has also been widely studied. Some authors described the apoptotic effect of HT in promyelocytic leukemia cells and colon adenocarcinoma cells [40,41]. HT also induces apoptosis in human breast MCF-7 cancer cells, but at higher concentrations tested than in the present study [38]. In our previous work, HT did not promote apoptosis in MDA-MB-231 [17]. Furthermore, SQ also did not exert pro-apoptotic activity in our previous studies [26]. Surprisingly, our results showed a pro-apoptotic activity of HT at 100 μM when it was combined with SQ, especially at high concentrations of SQ (10 and 100 μM) (Table 3). In addition, the amount of HT required to induce apoptosis in tumor cells was found to be lower when combined with SQ. As we observed in cell proliferation assay, it seems that SQ potentiates the activity of HT in some way.

It seems clear that SQ is, in some way, capable of enhancing the pro-apoptotic and antiproliferative activities of HT in human breast tumor cells. Since the existing research on the biological properties of VOO has focused on the study of isolated minority compounds, there are no references about how SQ could produce this interesting reinforcing effect. Nevertheless, Nakagawa et al. [42] observed that several anticancer agents such as Adriamycin, 5-fluorouracil, bleomycin and cis-dichlorodiamminoplatinum potentiated their antitumor activity when they were combined with SQ. The synergism of SQ with anticancer agents has also been described in vivo. SQ is able to potentiate the antitumor activity of 3-[(4-amino-2-methyl-5-pyrimidinyl) methyl]-1-(2-chloroethyl)-1-nitrosourea (ACNU) in a tumor murine model of leukemia, maybe by a modification of membrane permeability [43]. According to the authors, the synergism of SQ and antitumor agents may be due to the accumulation of anticancer drugs inside the cell because it interferes with drug efflux [43]. The anti-carcinogenic effects of SQ have been investigated together with Geranylgeranoic acid in murine models and it has been suggested that they could be due to the accumulation of FPP [21,22,23]. To our knowledge, only Bullon et al. [44] described the effects of HT and SQ in periodontal disease, where both compounds reversed gingival vascular damage after atherosclerotic diet, but there are no scientific data in cancer disease.

Unfortunately, the apoptotic and antiproliferative effects of HT–SQ combination only occur at high and non-physiological concentrations. However, as seen before, when HT is combined with SQ, the concentration of HT needed to maintain these antitumoral effects is dramatically reduced, compared to previous reports [17]. In fact, Yamaguchi et al. [40] claimed that the combination of anticancer agents with membrane-active molecules such as SQ could decrease the needed dose of anticancer drugs. Nowadays, the physiological concentration of HT after the ingestion of VOO remains unclear, but it could be between 10 and 100 μM [45]. In addition, SQ has the advantage that it is another minority compound that is naturally present in VOO. It is also present in animal cells, as a cholesterol biosynthesis intermediate. Therefore, the consumption of HT and SQ could be achieved by the Mediterranean diet, including VOO as the main fat.

## 5. Conclusions

The low risk of breast cancer associated with VOO intake may be a consequence of the synergistic activity of different minor compounds rather than the action of an individual compound. HT and SQ are two minor compounds present in VOO that possess antitumor activity on highly invasive breast tumor cells when acting together. However, we cannot extrapolate in vitro results to human populations; for this reason, further studies are needed to confirm its possible preventive role.

## Figures and Tables

**Figure 1 nutrients-14-00255-f001:**
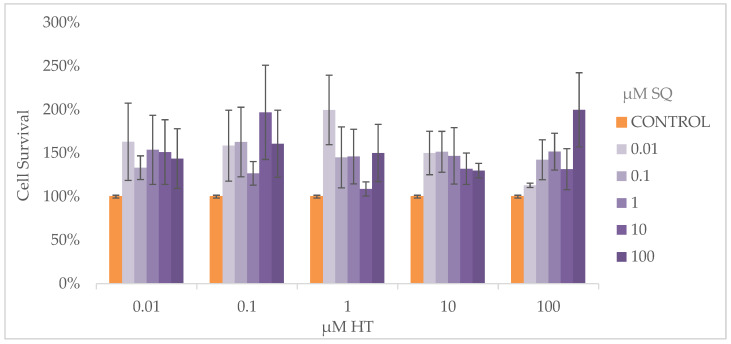
Cytotoxicity of HT–SQ from 0.01 μM to 100 μM in MDA-MB-231 cells. Data are represented as mean ± SEM of 3 independent experiments.

**Figure 2 nutrients-14-00255-f002:**
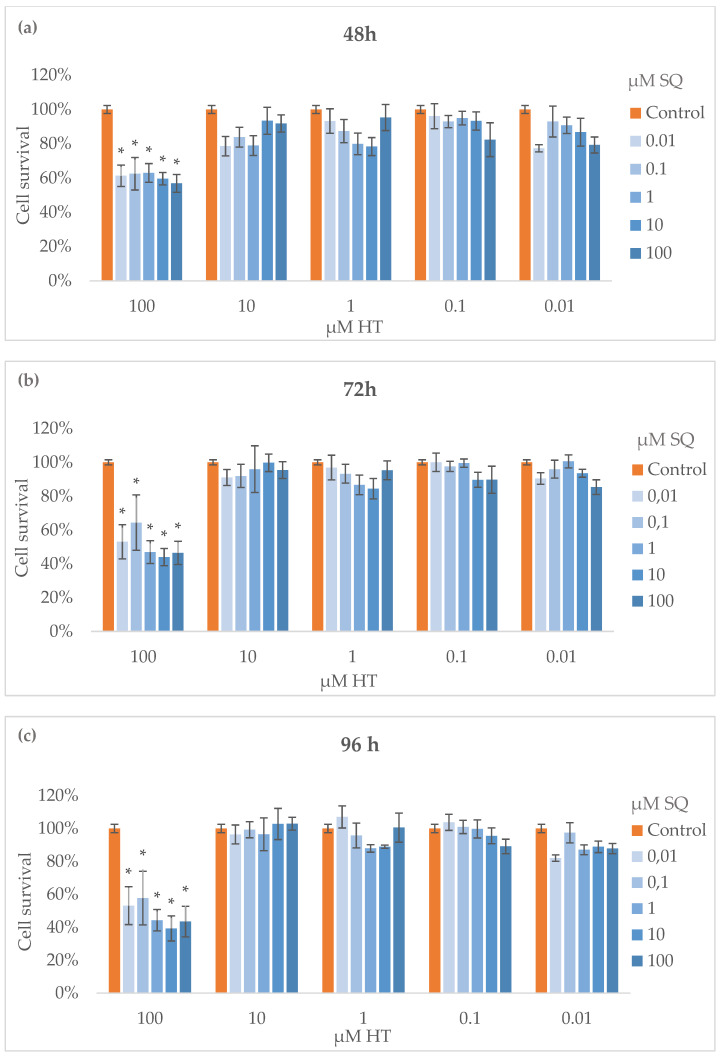
Cell proliferation after 48 h (**a**), 72 h (**b**) and 96 h (**c**) of a combination of HT–SQ from 0.01 μM to 100 μM in MDA-MB-231 cells. Data are represented as mean ± SEM of 3 independent experiments (* *p* < 0.05 compared to untreated control).

**Figure 3 nutrients-14-00255-f003:**
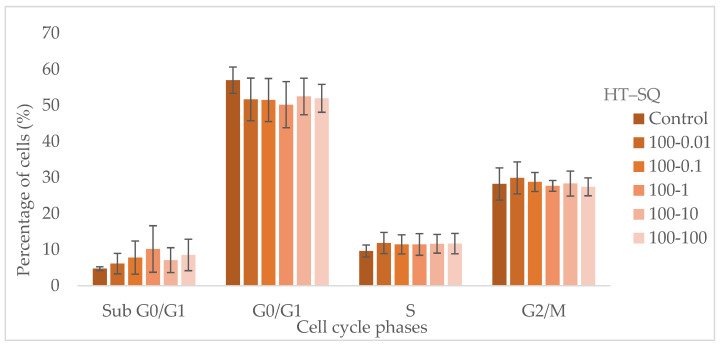
Distribution of cells in phases of the cell cycle for MDA-MB-231 cells after 24 h of treatment with HT at 100 μM combined with SQ at different concentrations (from 0.01 to 100 μM) and another 24 h of fresh medium. Data are represented as mean ± SEM of 3 independent experiments.

**Figure 4 nutrients-14-00255-f004:**
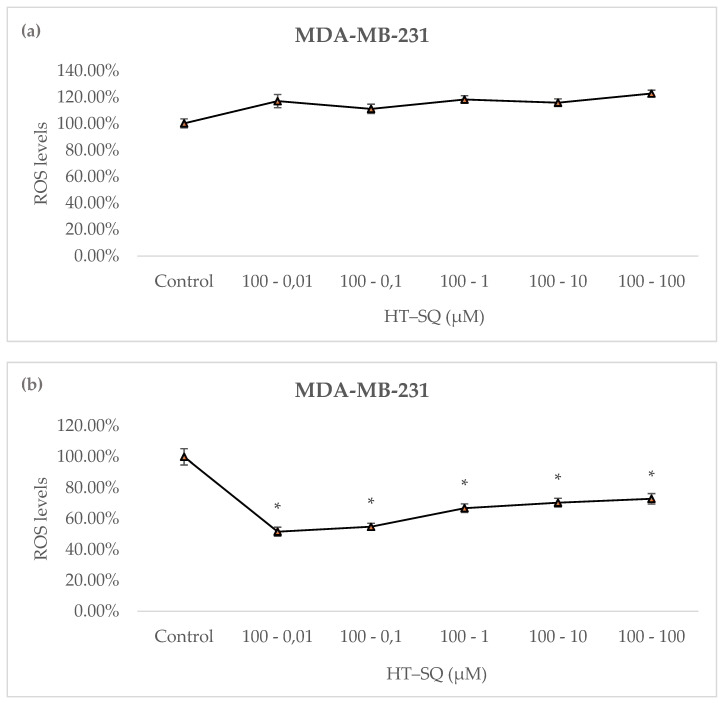
ROS levels represented in MDA-MB-231 cells in basal state (**a**) and with H_2_O_2_ burst (**b**) after treatment with HT at 100 μM, combined with SQ (from 0.01 to 100 μM) for 24 h, followed by another 24 h with fresh medium. Data are represented as mean ± SEM of 3 independent experiments (* *p* < 0.05).

**Table 1 nutrients-14-00255-t001:** Percentage of Free Radical Scavenging Activity of HT–SQ and α-tocopherol measured by the reduction in the DPPH Radical (% RSA at t = 60).

mol HT/mol DPPH–mol SQ/mol DPPH	HT–SQ (%)	mol α-tocopherol/mol DPPH	α-tocopherol (%)
**2–2**	43.66 ± 5.4	**2**	67.47 ± 3.26
**2–0.5**	41.12 ± 4.93	**0.5**	64.34 ± 3.71
**2–0.13**	52.16 ± 3.22	**0.13**	30.85 ± 10.06
**2–0.03**	44.62 ± 3.3	**0.03**	4.26 ± 8.8

Data are represented as mean ± SEM of 3 independent experiments at t = 60 min.

**Table 2 nutrients-14-00255-t002:** Percentage of Free Radical Scavenging Activity of the combination of both HT and SQ and Trolox^TM^ measured by ABTS assay (% RSA at t = 60). Values in the first and third columns represent HT and SQ concentrations, respectively. Values in the fifth column represent Trolox^TM^ concentrations.

HT–SQ (μM)	% RSA	Trolox^TM^ (μM)	% RSA
**100–100**	83.07 ± 1.27	**800**	85.92 ± 0.47
**100–10**	64.05 ± 8.57	**400**	76.79 ± 2.12
**100–1**	84.96 ± 0.68	**200**	43.41 ± 1.54
**100–0.1**	82.02 ± 1.76	**100**	24.82 ± 1.71
**100–0.01**	79.61 ± 2.07	**50**	14.75 ± 2
**10–100**	11.45 ± 2.43		
**10–10**	11.16 ± 2.25		
**10–1**	10.92 ± 2.73		
**10–0.1**	10.54 ± 2.43		
**10–0.01**	8.07 ± 2.39		

Data are represented as mean ± SEM of 3 independent experiments at t = 60 min.

**Table 3 nutrients-14-00255-t003:** Apoptosis of MDA-MB-231 cells treated with HT at 100 μM combined with SQ (from 0.01 to 100 μM) after 24 h of treatment plus 24 h of fresh medium.

HT–SQ (μM)	Live Cells	Apoptotic Cells	Necrotic Cells
Control	85.63 ± 3.06	11.38 ± 2.02	2.98 ± 1.14
100–0.01	76.13 ± 4.66	19.29 ± 2.61	4.58 ± 2.08
100–0.1	73.64 ± 1.99	22.52 ± 2.03	3.84 ± 1.67
100–1	75.32 ± 4.31	20.82 ± 2.18	3.87 ± 1.45
100–10	69.17 ± 5.87 *	25.68 ± 2.77 *	5.15 ± 2.12
100–100	69.34 ± 5.86 *	26.6 ± 2.98 *	4.07 ± 1.48

Data are represented as mean ± SEM of 3 independent experiments (* *p* < 0.05 compared to untreated control).

**Table 4 nutrients-14-00255-t004:** DNA damage in MDA-MB-231 cells in basal state (blue) and with H_2_O_2_ burst (orange) after treatment with HT at 100 μM combined with SQ (100, 10 and 1 μM) for 24 h followed by another 24 h with fresh medium.

		HT–SQ (μM)		
	Control	(100–100)	(100–10)	(100–1)
Basal	100 ± 5.79	92.88 ± 5.46	98.60 ± 6.54	129.45 ± 6.31 *
H_2_O_2_	100 ± 7.07	44.71 ± 3.64 *	34.17 ± 2.76 *	57.77 ± 6.60 *

Data are represented as mean ± SEM of 3 independent experiments expressed as percentage of Olive_TM (* *p* < 0.05 compared to untreated control).

## Data Availability

All data are available in the manuscript or upon request to the authors.

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
