# Peer review of "Synergistic Effect of Squalene and Hydroxytyrosol on Highly Invasive MDA-MB-231 Breast Cancer Cells"

_nutrients, 2022, doi:10.3390/nu14020255_

Round 1

Reviewer 1 Report

This is an original article on the association of the synagiristic effect of HT and squalene and the treatment of breast cancer. The manuscript was well prepared. Moreover, the results and their discussion are presented in good style. However, authors have a point that needs to be addressed.

1. Squalene is oxidized by monooxygenase in the early stages of metabolism.  However, its monooxygenase activity is highly suppressed by the accumulation of cholesterol (PMID:32170014).
Therefore, "when squalene in the diet is ingested, squalene-derived sterols accumulate in the cells and HMGCR is inhibited. As a result, the author's statement (L51-62) that FPP is depleted and prenylation, which promotes cancer," is suppressed can be a slight leap. I think it is necessary to simply touch on the possibility of FPP accumulating due to the oversupply of squalene. For example, papers such as PMID:30622150 report the biosynthesis of carcinogenic inhibitory lipids via GGPP after FPP accumulation by squale-statin. In addition, the most recent report is that the lipid is depleted in the elderly mouse liver, leading to carcinogenesis (PMID: 34564450).

Not limited to the above, I think that it would also be a better paper to discuss the possibility of carcinogenesis suppression and cancer cell death resulting from the accumulation of FPP associated with excess of squalene.

2. The quality of all graphs is poor. Please review all the graphs. For example, the font on the vertical axis, the method of% notation, significant figures, etc. are not unified throughout all the figures.

Author Response

Response to Reviewers

We would like to thank you for giving us the opportunity to submit this revised draft of the manuscript “Synergistic effect of squalene and hydroxytyrosol on highly invasive MDA-MB-231 breast cancer cells” for publication in Nutrients.

We are very grateful for these further suggestions you have offered and believe they have greatly improved the quality of our paper. Those changes are highlighted within the manuscript. We hope that, once the manuscript has been modified according to the reviewers' suggestions, it will be considered suitable for publication in Nutrients.

Reviewer 1

Comment: “This is an original article on the association of the synagiristic effect of HT and squalene and the treatment of breast cancer. The manuscript was well prepared. Moreover, the results and their discussion are presented in good style. However, authors have a point that needs to be addressed.”

Response: We appreciate the further time and effort dedicated to providing this extra feedback on our manuscript and are grateful for the insightful comments on and valuable improvements to our paper.  Below we have included your comments followed immediately by our responses.

Comment: “1. Squalene is oxidized by monooxygenase in the early stages of metabolism.  However, its monooxygenase activity is highly suppressed by the accumulation of cholesterol (PMID:32170014). Therefore, "when squalene in the diet is ingested, squalene-derived sterols accumulate in the cells and HMGCR is inhibited. As a result, the author's statement (L51-62) that FPP is depleted and prenylation, which promotes cancer," is suppressed can be a slight leap. I think it is necessary to simply touch on the possibility of FPP accumulating due to the oversupply of squalene. For example, papers such as PMID:30622150 report the biosynthesis of carcinogenic inhibitory lipids via GGPP after FPP accumulation by squale-statin. In addition, the most recent report is that the lipid is depleted in the elderly mouse liver, leading to carcinogenesis (PMID: 34564450). Not limited to the above, I think that it would also be a better paper to discuss the possibility of carcinogenesis suppression and cancer cell death resulting from the accumulation of FPP associated with excess of squalene.

Response: Thank you for your comment. Accordingly, a more in-depth explanation is assessed in lines 56-62 of the tracked version and has been introduced into the discussion section (lines 535-537).

Comment: “2. The quality of all graphs is poor. Please review all the graphs. For example, the font on the vertical axis, the method of % notation, significant figures, etc. are not unified throughout all the figures.”

Response: All graphics and tables have been revised and changed for obtaining a better quality.

Reviewer 2 Report

The title is appropriate to the article content.
The abstract adequately sums up the article.
Keywords are sufficient and appropriate.
In the introduction part, the reference search on the subject of the article has been done sufficiently.
The purpose of the article is clearly stated.
The study design of the article is appropriate.
Analysis methods to be applied to cell cultures are suitable.

Statistical analysis: The analysis method is suitable. However, the post hoc LSD test is the most prone to type 1 error. Therefore, if the variance of the data is homogeneous (if Levene test result is p>0.05), it is recommended to use Tukey's HSD or Bonferroni test; If the Levene test result is p<0.05, it is recommended to use Tamhane's-T2 test.

Tables and graphs are appropriate.

References are appropriate.

Author Response

Response to Reviewers

We would like to thank you for giving us the opportunity to submit this revised draft of the manuscript “Synergistic effect of squalene and hydroxytyrosol on highly invasive MDA-MB-231 breast cancer cells” for publication in Nutrients.

We are very grateful for these further suggestions you have offered and believe they have greatly improved the quality of our paper. Those changes are highlighted within the manuscript. We hope that, once the manuscript has been modified according to the reviewers' suggestions, it will be considered suitable for publication in Nutrients.

Reviewer 2

Comments: “The title is appropriate to the article content.
The abstract adequately sums up the article.
Keywords are sufficient and appropriate.
In the introduction part, the reference search on the subject of the article has been done sufficiently.
The purpose of the article is clearly stated.
The study design of the article is appropriate.
Analysis methods to be applied to cell cultures are suitable.”

Response: We appreciate the further time and effort dedicated to providing this extra feedback on our manuscript and are grateful for the insightful comments on and valuable improvements to our paper.  Below we have included your comments followed immediately by our responses.

Comments: Statistical analysis: The analysis method is suitable. However, the post hoc LSD test is the most prone to type 1 error. Therefore, if the variance of the data is homogeneous (if Levene test result is p>0.05), it is recommended to use Tukey's HSD or Bonferroni test; If the Levene test result is p<0.05, it is recommended to use Tamhane's-T2 test.

Tables and graphs are appropriate.

References are appropriate.”

Response: Thank you for your comment. Statisticians call the risk, or probability, of making a Type I error "alpha," aka "significance level." In other words, it's our willingness to risk rejecting the null when it's true. Alpha is commonly set at 0.05, which is a 5 percent chance of rejecting the null when it is true. Unless we know that the LSD test can be prone to type 1 error, due to our kind of data we decide to apply this analysis method. We believe that, due to the characteristics of our study, the LSD test showed the best statistical adjustment to our data.